# Closed-form feedback-free learning with forward projection

Robert O'Shea [1] & Bipin Rajendran [1,2]

State-of-the-art backpropagation-free learning methods employ local error feedback to direct iterative optimisation via gradient descent. Here, we examine the more restrictive setting where retrograde communication from neuronal outputs is unavailable for pre-synaptic weight optimisation. We propose **Forward Projection** (FP), a randomised closed-form training method requiring only a single forward pass over the dataset without retrograde communication. FP generates target values for pre-activation membrane potentials through randomised nonlinear projections of pre-synaptic inputs and labels. Local loss functions are optimised using closed-form regression without feedback from downstream layers. A key advantage is interpretability: membrane potentials in FP-trained networks encode information interpretable layer-wise as label predictions. Across several biomedical datasets, FP achieves generalisation comparable to gradient descent-based local learning methods while requiring only a single forward propagation step, yielding significant training speedup. In few-shot learning tasks, FP produces more generalisable models than backpropagation-optimised alternatives, with local interpretation functions successfully identifying clinically salient diagnostic features.

A core task in neural network training is synaptic "credit assignment" for error in downstream layers[1-3]. Although backpropagation has been established as the standard approach to this problem, its biological plausibility has been questioned[3-5]. Backpropagation requires bidirectional synaptic communication, which is incompatible with the unidirectional transmission of neural action potentials[3]. Consequently, backpropagation would require either symmetric neural connectivity or a parallel retrograde network for error feedback to earlier layers[3,6,7]. The backward pass traverses layers in reverse order of activation, leading to temporal discordance between forward and backward operations and necessitating storage of hidden activations. Furthermore, gradient descent requires hidden neural activations to be differentiable throughout. To approximate backpropagation in a more biologically plausible way, approaches such as Predictive Coding[5,8] and Difference Target Propagation[1] seek to align forward neural activity with a backward network that mirrors the forward architecture, effectively implementing an inverse model. Various strategies have

been proposed to address the credit assignment problem with reduced retrograde communication requirements. Auxiliary loss functions computed on the activations of individual layers have been proposed to shorten the backward pass[9-12]. A prominent example is the layer-wise greedy optimisation method that aims to minimise the costs of backpropagation by applying local supervision (LS) at each layer[9,13]. Extensions of this approach include deep continuous local learning[10] and single-layer updating[14]. Hebbian learning rules using asymmetric feedback weights have been explored, allowing for independent updates for forward and feedback pathways[15].

Recent approaches have also explored using two forward passes to facilitate communication between upstream and downstream neurons[16-19]. The "Forward-Forward" (FF) learning algorithm[16] is an approach in which data and label hypotheses are combined as inputs, with optimisation seeking to upregulate neural response to correctly labelled inputs and subdue responses to spuriously labelled inputs. However, inference under the original Forward-Forward algorithm

[1]Centre for Intelligent Information Processing Systems, Department of Engineering, King's College London, London, UK. [2]Present address: Institute for Intelligent Networked Systems, Northeastern University London, E1 8PH London, United Kingdom. ✉e-mail: k1930297@kcl.ac.uk; bipin.rajendran@kcl.ac.uk

requires a forward pass for each hypothesised label, presenting issues for tasks with large label spaces[19]. The "PEPITA" algorithm employed a preliminary forward pass to predict a label, which is used to generate spurious data-label instances for training[17].

The central issue with local learning methods in deep neural networks is that the optimal activity of hidden neurons is unknown during training, preventing direct observation of local error. Predefining activity targets for hidden neurons allows for heuristic local optimisation. However, target definition strategies are a topic of ongoing research[20]. Techniques such as local-representation alignment and target propagation introduce target values for hidden activations using limited retrograde communication from downstream neurons[1,11,20–22]. Alternatively, target activities can be set as fixed random label projections computed during the forward pass, thereby permitting direct measurement of the error during the forward pass[11]. However, this approach may lead to highly correlated neuronal activity – a problem known as informational collapse[13]. Although additive noise permits maximal decorrelation of target potentials[1], it is uninformative with respect to the label, potentially impeding model fitting. Approaches such as random neural network features[23] forgo optimisation of hidden nodes entirely, instead projecting inputs to a random high-dimensional non-linear feature space. However, random feature layers require exponential scaling with respect to the input dimension to support downstream learning[24].

The "Predictive Coding" paradigm provides a perspective on the collective functionality of intermediate neurons, proposing that local neural learning processes optimise the prediction of presynaptic neural inputs, minimising the "surprise" of out-of-distribution stimuli[5]. Approaches such as the "Difference Target Propagation"[1] reframe network layers as a series of autoencoders, where intermediate activations represent a series of encodings, transitioning from input information to label information. The issue of informational collapse in locally supervised SGD-trained models has been addressed by optimising the retention of information on the pre-synaptic activity[13]. "Prospective configuration" of target activities reformulates learning under the presumption that idealised adjustments to neuronal activity should be generated via energy minimisation before synaptic adjustment[20]. Synaptic weights may be modified to realise the predetermined activities in response to the given input stimuli[20], thereby transitioning from input information to label information through model layers. "Local neural synchronisation" generates target neural activity for hidden layers by projecting neuronal activity onto periodic basis vectors representative of class labels[25].

Going beyond these approaches, we propose the use of random nonlinear projections of both pre-synaptic inputs and target labels to generate local target activities in the forward pass. The objective of this approach is to develop a neural network training algorithm that requires no retrograde communication. Closed-form regression techniques are applicable in this setting, permitting single-step layer weight computation without error feedback. Thus, weight parameters are determined without backward communication from neuronal outputs or downstream layers. Further advantages of this method include the direct interpretability of hidden neurons with respect to local label predictions and stability in the few-shot setting.

## Results

We consider a dataset $\mathcal{D} = \{(\mathbf{x}_i, \mathbf{y}_i)\}_{i=1}^N$ from the joint distribution $(X, Y)$. The task is to learn a feed-forward neural network function mapping $X \rightarrow Y$. The model has $L$ layers, with dimensions $m_0, ..., m_L$, and activations $\mathbf{a}_0, ..., \mathbf{a}_L$, where $\mathbf{a}_0 = \mathbf{x}$ and $\mathbf{a}_L = \mathbf{y}$. Each layer is equipped with weights $\mathbf{W}_l$ to generate membrane potentials $\mathbf{z}_l = \mathbf{a}_{l-1}\mathbf{W}_l$, and activation function $f_l : \mathbb{R} \rightarrow \mathbb{R}$ to generate neuronal outputs $\mathbf{a}_l = f_l(\mathbf{z}_l)$. The model prediction is defined as $\hat{\mathbf{y}} = f_L(\mathbf{W}_L(\ldots f_1(\mathbf{W}_1\mathbf{x})))$.

### Forward projection (FP)

To generate target membrane potentials for hidden neurons, we present the Forward Projection (FP) algorithm (Fig. 1A). We propose to combine pre-synaptic inputs and labels using random non-linear projections to generate targets (Fig. 1B). For each training sample, the target potential $\tilde{\mathbf{z}}_l \in \mathbb{R}^{1 \times m_l}$ is generated from pre-synaptic inputs $\mathbf{a}_{l-1} \in \mathbb{R}^{1 \times m_{l-1}}$ and labels $\mathbf{y} \in \mathbb{R}^{1 \times m_l}$ using fixed random projection matrices $\mathbf{Q}_l \in \mathbb{R}^{m_{l-1} \times m_l}$ and $\mathbf{U}_l \in \mathbb{R}^{m_l \times m_l}$ such that

$$\tilde{\mathbf{z}}_l = g_l(\mathbf{a}_{l-1}\mathbf{Q}_l) + g_l(\mathbf{y}\mathbf{U}_l), \tag{1}$$

where $g_l : \mathbb{R} \rightarrow \mathbb{R}$ is an element-wise non-linear transformation. It is noted here that $g_l$ is not necessarily equal to $f_l$, the neural activation function. $\mathbf{Q}_l$ and $\mathbf{U}_l$ are fixed linear projections drawn from random Gaussian distributions, which are pre-defined before training. The target potential $\tilde{\mathbf{z}}_l$ allows a local auxiliary loss function to be defined with respect to the actual membrane potential, $\mathbf{z}_l$, realised during the forward pass. The randomised combination of pre-synaptic inputs with the target label is inspired by the high-dimensional computing paradigm[26], where fixed random projections are employed to encode information from multiple vector inputs, approximately preserving relative distances according to the Johnson-Lindenstrauss lemma[26]. Accordingly, it is noted that $\tilde{\mathbf{z}}_l$ approximately encodes both $\mathbf{a}_{l-1}$ and $\mathbf{y}$ (see section S.8 in SI). By generating target potentials in a forward manner, the mix of information encoded in each layer is expected to transition incrementally from predominantly input information in early layers to label information in later layers. We propose to define a local loss function $\mathcal{L}_l := \| \mathbf{Z}_l - \tilde{\mathbf{Z}}_l \|$, which may be employed as an objective to optimise $\mathbf{W}_l$. Synaptic weights for each layer can be computed in a closed-form forward manner, using ridge regression (Fig. 1C), such that

$$\mathbf{W}_l := (\mathbf{A}_{l-1}^\top \mathbf{A}_{l-1} + \lambda \mathbf{I})^{-1}(\mathbf{A}_{l-1}^\top \tilde{\mathbf{Z}}_l). \tag{2}$$

Here, $\mathbf{A}_{l-1} \in \mathbb{R}^{N \times m_{l-1}}$ and $\tilde{\mathbf{Z}}_l \in \mathbb{R}^{N \times m_l}$ are matrices of pre-synaptic activities and target potentials, respectively, collected over the $N$ samples in $\mathcal{D}$. $\lambda$ is a regularisation term, and $\mathbf{I}$ is the identity matrix. Observe that the $\mathbf{A}_{l-1}^\top \mathbf{A}_{l-1}$ and $\mathbf{A}_{l-1}^\top \tilde{\mathbf{Z}}_l$ terms in (2) may be computed sequentially over instances $\{\mathbf{a}_{1,l-1}, \ldots, \mathbf{a}_{n,l-1}\} \subset \mathbb{R}^{1 \times m_{l-1}}$, and $\{\tilde{\mathbf{z}}_{1,l}, \ldots, \tilde{\mathbf{z}}_{N,l}\} \subset \mathbb{R}^{1 \times m_l}$ (see section S.7 in SI). Therefore, memory requirements are independent of $N$, depending only on $m_{l-1}$ and $m_l$. Note, weights are computed once only, after the single training epoch. The approach of encoding information on both inputs and labels in the hidden layers is inspired by the Predictive Coding and Target Propagation paradigms[1,5,9]. However, unlike previous approaches, no backward communication is required from neuronal outputs to presynaptic inputs to achieve the FP fit. As a consequence of this fitting approach, neural membrane potentials in hidden layers of FP-trained models may be interpreted as label predictions (Fig. 1C).

### Forward projection performance

Forward Projection was compared to other local learning methods, including random features (RF)[24], Local Supervision (LS)[9], and Forward-Forward (FF)[16] in multiple tasks using equivalent model architectures (Table 1). The Fashion MNIST (FMNIST) image classification task was modelled on a multi-layer perceptron (MLP). Forward Projection achieved higher test accuracy than other local learning methods in this task, approaching the performance of the backpropagation reference standard. Two large-scale biomedical sequence modelling tasks were also evaluated with one-dimensional convolutional neural network (1D-CNN) architectures. The Promoters task required the identification of human non-TATA promoters, a class of gene promoter regions that increase transcription of DNA sequences[27,28]. Forward projection yielded higher test performance than all other local learning methods in this task. The PTBXL-MI task[29]

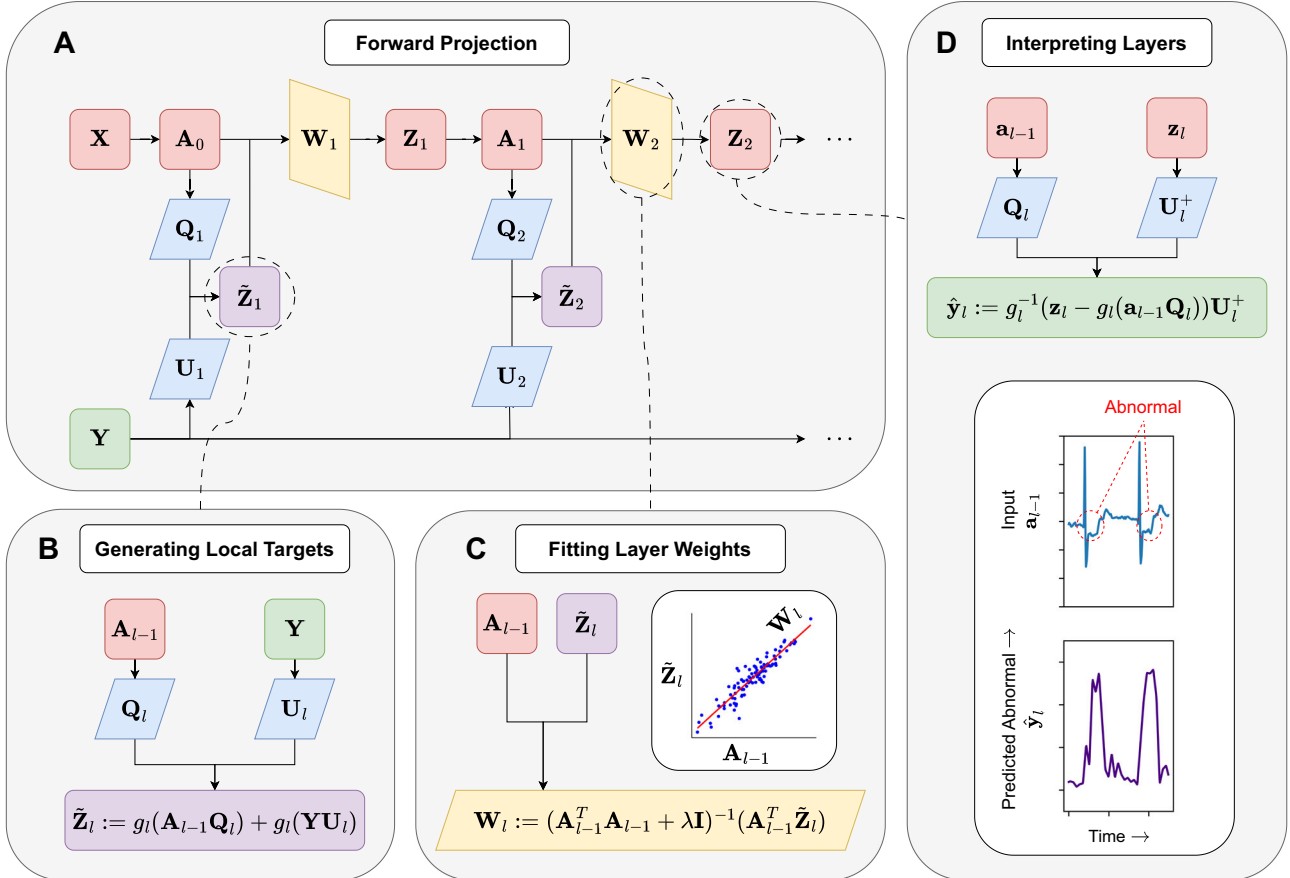

**Fig. 1 | Graphical overview of forward projection. A** Forward projection algorithm for fitting layer weights $\mathbf{W}_1, \ldots, \mathbf{W}_l$ to model labels $\mathbf{Y}$ from data $\mathbf{X}$. **B** Procedure for generating the $l$-th layer target potentials $\tilde{\mathbf{Z}}_l$. Pre-synaptic inputs $\mathbf{A}_{l-1}$ and labels $\mathbf{Y}$ are projected with fixed matrices $\mathbf{Q}_l$ and $\mathbf{U}_l$, respectively, before applying non-linearity $g_l$. **C** Optimising $\mathbf{W}_l$ to predict $\tilde{\mathbf{Z}}_l$ from $\mathbf{A}_{l-1}$ by ridge regression with penalty $\lambda$. **D** Interpreting membrane potentials $\mathbf{z}_l$ as a local label prediction $\hat{\mathbf{y}}_l$ given pre-synaptic inputs $\mathbf{a}_{l-1}$ and projection matrices $\mathbf{Q}_l$ and $\mathbf{U}_l^+$, where $\mathbf{U}_l^+$ is the pseudo-inverse of $\mathbf{U}_l$.

required diagnosis of myocardial infarction (MI), a heart condition commonly known as "heart attack", from 12-channel electro-cardiogram (ECG) recordings. FP and LS performed comparably in this task. FP also outperformed all other methods in the optimisation of models with more complex neuronal activations such as modulo and polynomial activation functions (see section S.14 in SI). Back-propagation performance benchmarks on the PTBXL-MI and Pro-moters tasks were consistent with previous studies[28,30]. For the CIFAR10 classification task without data augmentation, two-dimensional convolutional neural networks (2D-CNNs) trained by FP underfitted, but still outperformed all other local learning methods, whereas standard backpropagation overfitted the model. FP was implemented in a transformer architecture to discriminate the first two classes of the CIFAR10 dataset ("CIFAR2"), outperforming RF.

**Alternative feedback-free approaches.** To assess the value of the FP target generation function, we compared the performance of closed-form regression models fitted to targets generated by alternative functions, including simple label projection ($\tilde{\mathbf{Z}}_l := \mathbf{y}\mathbf{U}_l$) and label projection with additive noise ($\tilde{\mathbf{Z}}_l := \mathbf{y}\mathbf{U}_l + \mathbf{E}$). To evaluate the cap-ability of feedback-free training methods to handle information "bot-tlenecks", they were applied to optimise MLP architectures for FMNIST classification. MLPs were generated with 1000 hidden neurons in the first and second layers ($m_1 = m_2 = 1000$), and $m_3 \in \{100, 200, 400, 800\}$ neurons in the final hidden layer. FP outperformed other feedback-free approaches (Fig. 2A). The performance of RF deteriorated in models with small penultimate layers, as relevant information was less likely to be represented by random projection[24]. The performance of simple

label projection deteriorated in models with large penultimate layers, a result that may be attributable to rank deficiency (see section S.6 in SI). Noisy label projection maintained steady performance, but at a lower level than FP.

**Few-shot learning**

"Few-shot" learning is a constrained learning scenario in which the number of data samples available for training is small. High-dimensional data, such as images, pose a challenge for few-shot learning, as many spurious features may exist. Thus, successful few-shot training methods must select generalisable features in the pre-sence of these confounders. Few-shot learning was assessed in image classification tasks using a 2D-CNN architecture. The optical coherence tomography task (OCT)[31] required that the model discriminate images of healthy retinas from those with choroid neovascularization, a pathology that affects the eye and manifests mainly as abnormal growth of blood vessels behind the retina. The paediatric chest X-ray (CXR) task[31] required the model to discriminate between images of viral pneumonia, bacterial pneumonia, or healthy controls. The CIFAR2 task required the model to discriminate between the first two CIFAR10 classes (aeroplane and automobile). Few-shot training datasets were generated by subsampling, with $N \in \{5, 10, 15, 20, 30, 40, 50\}$ training examples from each class for OCT and CXR tasks, and $N \in \{25, 50, 75, 100\}$ training examples per class for the CIFAR2 task. Data augmentation was not employed. Model generalisability was assessed on all test data (CXR: $N_{\text{test}} = 431$; OCT: $N_{\text{test}} = 327$; CIFAR2: $N_{\text{test}} = 2000$). Forward Projection-trained models demonstrated the greatest few-shot generalisability in CXR (Fig. 2B) and OCT (Fig. 2C),

**Table 1 | Test performance of various learning methods across different datasets**

| Dataset | Method Metric | FP (ours) | RF | LS | FF | PC | DTP | BP (ref.) |
|---|---|---|---|---|---|---|---|---|
| FMNIST MLP | AUC | 98.3 ±0.0 | 98.0 ±0.0 | 98.7 ±0.1 | 98.5 ±0.1 | 98.2 ±0.1 | 97.8 ±0.2 | 99.0 ±0.0 |
| | Acc | 86.3 ±0.1 | 84.0 ±0.2 | 83.1 ±2.0 | 85.6 ±0.8 | 83.0 ±0.4 | 79.3 ±2.0 | 88.4 ±0.3 |
| Promoters 1D-CNN | AUC | 88.7 ±0.5 | 80.0 ±1.1 | 86.6 ±1.1 | 83.5 ±0.7 | 79.7 ±1.7 | 78.8 ±0.8 | 94.1 ±0.3 |
| | Acc | 81.8 ±0.5 | 72.2 ±1.4 | 79.7 ±1.2 | 75.3 ±3.6 | 70.5 ±3.6 | 69.3 ±3.3 | 87.4 ±0.7 |
| PTBXL-MI 1D-CNN | AUC | 95.5 ±0.5 | 94.7 ±1.8 | 97.3 ±0.3 | 89.3 ±3.8 | 86.0 ±0.9 | 86.4 ±2.2 | 99.3 ±0.0 |
| | Acc | 86.5 ±1.1 | 83.4 ±2.1 | 86.1 ±4.0 | 69.4 ±8.2 | 64.2 ±5.0 | 70.1 ±5.8 | 94.9 ±0.6 |
| CIFAR2 2D-ViT | AUC | 91.5 ±0.5 | 83.1 ±0.5 | | | | | 96.1 ±0.4 |
| | Acc | 83.5 ±0.5 | 75.9 ±0.6 | | | | | 89.2 ±0.7 |
| CIFAR10 2D-CNN | AUC | 85.1 ±0.2 | 81.4 ±0.5 | 87.7 ±0.5 | 57.9 ±1.7 | 74.5 ±0.3 | 64.5 ±2.3 | 89.8 ±0.7 |
| | Acc | 48.8 ±0.3 | 40.6 ±1.0 | 48.4 ±1.1 | 13.1 ± 1.9 | 28.5 ±0.9 | 15.3 ±3.0 | 54.8 ± 1.5 |

The FMNIST dataset was modelled with a multi-layer perceptron; Promoters and PTBXL-MI datasets were modelled using 1D convolutional neural networks (1D-CNNs); CIFAR2 (first two CIFAR10 classes only) was modelled using vision transformers, and CIFAR10 was modelled with 2D-CNNs.

*BP* backpropagation, *DTP* difference target propagation, *FF* forward-forward, *FP* forward projection, *LS* local supervision, *PC* predictive coding, *RF* random features.

outperforming all other methods, including backpropagation, in all the tasks with $N \leq 40$ training samples. In the CIFAR2 few-shot task, Forward Projection achieved the best performance of any local learning method in experiments with $N \in \{50, 75, 100\}$, being outperformed only by backpropagation. With as few as $N = 10$ training samples, Forward Projection fitted discriminative models for the classification of OCT (test AUC: 84.5 ± 5.8) and CXR (test AUC: 76.6 ± 5.8). CXR and OCT datasets highlighted two distinct vulnerabilities of backpropagation training in few-shot conditions. In the $N = 5$ setting on OCT, backpropagation overfitted the training samples (Train AUC: 86.8 ± 13.2; Test AUC: 61.8 ± 17.4), as models integrated noise into decision functions. On the other hand, backpropagation failed to achieve adequate model fitting in the $N = 10$ setting on CXR (Train AUC: 69.0 ± 8.1; Test AUC: 59.7 ± 6.2). Random features performed comparably to backpropagation in tasks with the fewest training samples, but performance improved only slightly with larger sample sizes. Random Features models could not overfit within convolutional layers, as these contained no free parameters. However, Random Features had a limited capacity to learn structural features in larger training samples. The improvement that Forward Projection provided over Random Features is therefore attributable to structural feature learning within hidden convolutional layers. Label Projection and Noisy Label Projection failed to converge in few-shot training (training AUC ≈ 0.50). Neither Local Supervision nor Forward-Forward training matched the baseline generalisability of Random Features in OCT and CXR tasks; however, Local Supervision achieved comparable test discrimination to Forward Projection in the CIFAR2 task when $N \geq 75$. Predictive coding and Difference Target Propagation yielded uninformative models in few-shot learning tasks (Supplementary Fig. S3). Few-shot performance is tabulated in Supplemental Table S2.

**Feature interpretability**

Explainability is a central issue with backpropagation-based learning, as relationships between hidden activations and model predictions may be non-monotonic, complicating the interpretation of hidden neural activities[32]. An important advantage of Forward Projection is the interpretability of hidden neuron activity with respect to label predictions (see section S.8 in SI). Assuming $\mathbf{z}_l \approx \tilde{\mathbf{z}}_l$, neural potentials may be interpreted as a local label prediction $\hat{\mathbf{y}}_l$ (Fig. 1D), such that

$$\hat{\mathbf{y}}_l := g_l^{-1}(\tilde{\mathbf{z}}_l - g_l(\mathbf{a}_{l-1}\mathbf{Q}_l))\mathbf{U}_l^+ . \qquad (3)$$

Here, $\mathbf{U}_l^+$ is the Moore-Penrose generalised inverse of the label projection matrix. Likewise, pre-synaptic inputs are encoded in neural pre-activation potentials, with an analogous reconstruction function

(see section S.8 in SI). It is noted that (3) may be uninformative if $\frac{\|\mathbf{z}_l - \tilde{\mathbf{z}}_l\|}{\|\mathbf{z}_l\|}$ is large − i.e., if $\mathbf{W}_l$ did not achieve a good fit. Measurement of local error in training data may provide insights into the reliability of (3) during inference. Interpretation of pre-activation potentials $\mathbf{z}_l$ is simplified by the selection of bijective functions for $g_l$ so that $g_l^{-1}$ exists everywhere. In practice, a surrogate approximation to the functional inverse was observed to suffice in our experiments; for example, $g_l^{-1}(\cdot) \approx \tanh(\cdot)$ was employed as a surrogate inverse for $g_l(\cdot) = \text{sign}(\cdot)$. In our experiments, hidden neurons of models fitted with Forward Projection were interpretable as label predictions. In the FMNIST task, test accuracy of layer explanations was observed to improve between early layers and subsequent layers (Supplementary Fig. S2-A), demonstrating progressive learning. In the PTBXL-MI task, applying the surrogate layer explanation function (described in equation (3)) to the convolutional layers identified various clinically salient features for diagnosing myocardial infarction (MI). MI, a clinical condition characterised by damage to heart muscles due to poor blood flow, may manifest in ECG data with various electrophysiological abnormalities. Consequently, the model must learn several distinct pathological waveform features, including elevation of the "ST" segment, or inversion of the "T" wave. Fig. 3A shows the layer explanations as a function of time in four patients, three of whom were diagnosed with myocardial infarction. Patient A demonstrates ST-segment depression in lead II, which is temporally consistent with peaks in the model explanation functions at each layer. Likewise, the model explanation functions peak during ST-segment elevation in Patient B and during QRS widening and T-wave inversion in Patient C. In contrast, the model explanation function is near zero in Patient D, who had normal ECG morphology. Forward Projection model explanations derived using (3) from the sixth convolutional layer ($\hat{y}_6$) were compared with GradCAM outputs of backpropagation-trained networks. The first 15 MI-positive test instances in the PTBXL dataset were annotated by a medical doctor to segment the diagnostically relevant subsequences. Layer explanations derived by Forward Projection achieved similar discrimination performance (AUC: 0.61 ± 0.14, AUPR: 0.52 ± 0.19) to GradCAM (AUC: 0.59 ± 0.12, AUPR: 0.55 ± 0.13) (Fig. 3B).

Choroid neovascularization (CNV) is the growth of abnormal blood vessels behind the retina due to diseases such as age-related macular degeneration[33]. In OCT images, CNV may be represented by various image features, including hyper-reflective dots and detachment of the retinal pigment epithelium[34]. In the OCT task, the model's layer explanation functions identified regions of interest related to CNV (Fig. 4). 2D-CNN models trained with only 100 instances per class

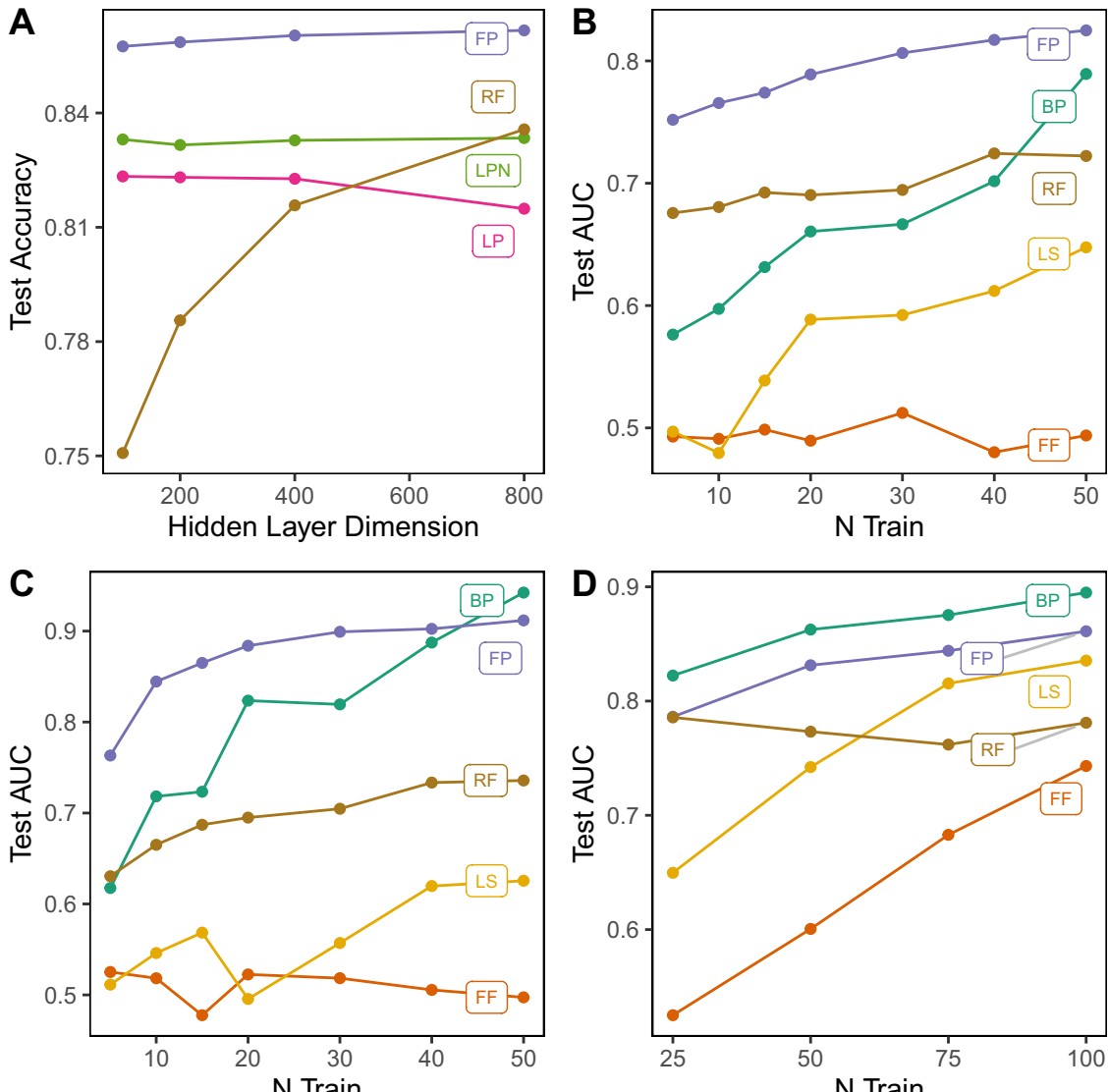

**Fig. 2 | Performance of Forward Projection with backpropagation and local learning approaches. A** Comparison of feedback-free fitting methods on FMNIST. MLP architectures had 1000 neurons in the first and second layers and 100, 200, 400 or 800 neurons in the final hidden layer. **B**–**D** Test performance of few-shot trained 2D-CNN models. Mean test AUC is reported over 50 few-shot training experiments. **B** Chest X-ray (CXR) task. **C** Optical Coherence Tomography (OCT) task. **D** CIFAR2 task in which models were required to classify the first two classes (aeroplane and automobile). Models were fitted with $N \in \{5, 10, 15, 20, 30, 40, 50\}$ training samples from each class in CXR and OCT tasks and $N \in \{25, 50, 75, 100\}$ samples per class for the CIFAR2 task. Predictive Coding and Difference Target Propagation are plotted separately in Supplementary Fig. S3. BP backpropagation, FF Forward-Forward, FP Forward Projection, LS Local Supervision, RF Random Features.

learned to localise fine-grained CNV features, including retinal/sub-retinal fluid (Patients A-C), hard exudates (Patient B), and fibrosis (Patient C).

## Training complexity

We now analyse the complexity of training with the Forward Projection algorithm for a classification task by estimating the storage and computational requirements for a densely connected $m \times m$ hidden layer. We consider a dataset of $N$ training samples, with label dimension $m_L$. Note that FP model weights can be obtained in one pass over the $N$ training samples, while all other methods require each training sample to be fed to the network for several epochs (denoted $N_e$). The training procedure for each layer is presented in Supplementary Fig. S1.

The memory requirement for Forward Projection is $\mathcal{O}(m^2)$ for the layer weights, $\mathcal{O}(m^2)$ for the **Q** matrix, and $\mathcal{O}(mm_L)$ for the **U** matrix. As the $\mathbf{A}_{l-1}^\top \mathbf{A}_{l-1}$ and $\mathbf{A}_{l-1}^\top \widetilde{\mathbf{Z}}_l$ terms in (2) can be accumulated sequentially over data batches (See section S.7 in SI), two $m \times m$ matrices suffice for their storage, thereby avoiding storage of the $N \times m$ matrices $\mathbf{A}_{l-1}$ and $\widetilde{\mathbf{Z}}_l$. As with all methods, Forward Projection requires $\mathcal{O}(m^2)$ multiply-and-accumulate (MAC) operations to calculate the activations for the downstream layer in the forward pass. To generate target potentials during the forward pass, Forward Projection also requires $\mathcal{O}(m^2)$ additional MAC operations to project through $Q$ and $\mathcal{O}(mm_L)$ MAC operations to project through **U**. Computation of the $\mathbf{A}_{l-1}^\top \mathbf{A}_{l-1}$ and $\mathbf{A}_{l-1}^\top \widetilde{\mathbf{Z}}_l$ terms requires $\mathcal{O}(Nm^2)$ MAC operations each. After all samples have been observed, computation of model weights using equation (2) requires $\mathcal{O}(m^3)$ operations to invert the $\mathbf{A}_{l-1}^\top \mathbf{A}_{l-1}$ term and $\mathcal{O}(m^3)$ operations for matrix multiplication to complete the regression; however, this happens only once for each layer. Furthermore, Forward Projection requires no backward pass.

For backpropagation, $\mathcal{O}(m^2)$ memory is required for storing layer weights, $\mathcal{O}(m^2)$ for accumulated gradients, and $\mathcal{O}(m)$ for activations.

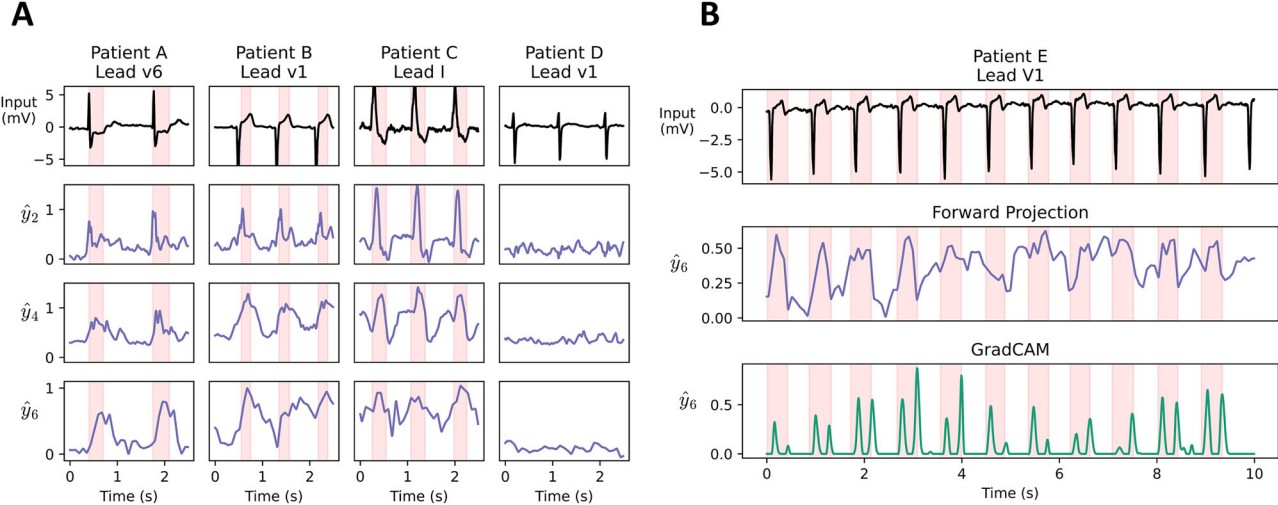

**Fig. 3 | Forward Projection layer interpretations for electrocardiogram analysis. A** Visualisation of layer explanations over time in a 1D-convolutional neural network trained by Forward Projection to detect myocardial infarction (MI) in electrocardiograms (ECGs) from PTBXL data. Patients A, B, C, E (diagnosed with MI) and patient D (no disease) were extracted from test data. Explanations were extracted from the second, fourth and sixth convolutional layers ($\hat{y}_2, \hat{y}_4, \hat{y}_6$) using equation (3). Explanations increase with MI features (highlighted in red), including ST-segment depression (Patient A), ST-segment elevation (Patient B) and QRS widening with T-wave inversion (Patient C). **B** Comparison of Forward Projection with GradCAM. Top: ECG data from Patient E (diagnosed MI), showing ST-segment elevation. Middle: Sixth convolutional layer explanation ($\hat{y}_6$) from a model trained by Forward Projection. Below: GradCAM output for the sixth convolutional layer of a model trained by backpropagation.

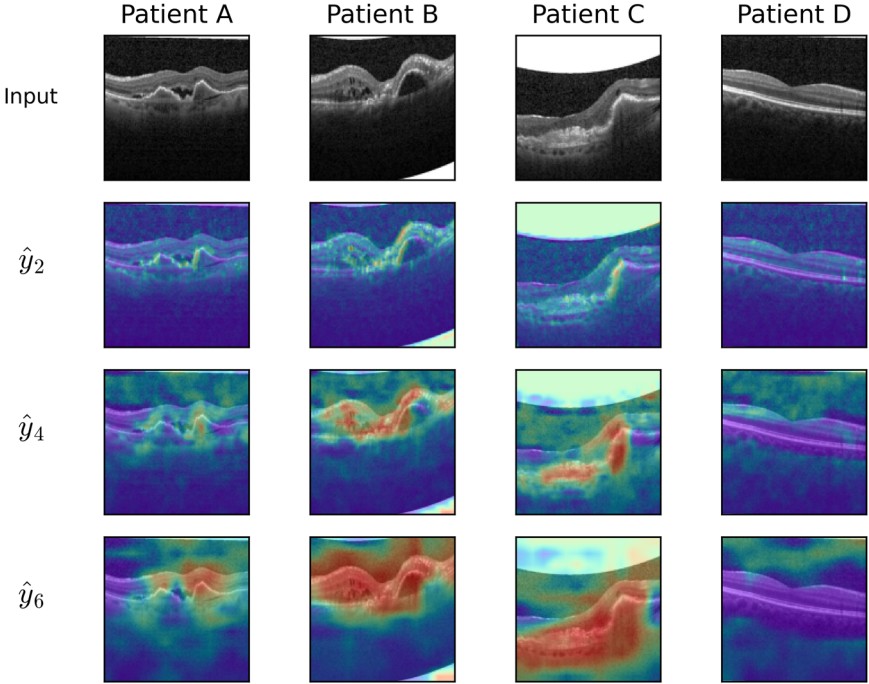

**Fig. 4 | Visualisation of layer explanations over space in 2D-CNNs trained by Forward Projection to detect choroid neovascularization (CNV) in the OCT task.** Ensemble average of five models shown. Patients A–C (diagnosed with CNV) and patient D (no disease) were extracted from test data. Explanations were extracted from the second, fourth and sixth convolutional layers ($\hat{y}_2, \hat{y}_4, \hat{y}_6$), using (3). CNV heat-maps demonstrate high values (red) over CNV features, including retinal/subretinal fluid (Patients A–C) and hard exudates (Patient B), and fibrosis (Patient C), with low values (blue) over the healthy retina (Patient D).

Here, the computation requirements scale as $\mathcal{O}(N_e m^2)$ MACs each for the forward pass, backward pass, and weight update calculations. Hence, in the typical setting where $m \gg m_L$, we note that Forward Projection and backpropagation have similar memory requirements, scaling as $\mathcal{O}(m^2)$. Notably, compute scales as $\mathcal{O}(m^3)$ for FP versus $\mathcal{O}(N_e m^2)$ for BP. For example, training a dense hidden layer with 1000 inputs and 1000 outputs on the FMNIST dataset ($N = 60,000, m_L = 10$) by Forward Projection requires a total of $1.2 \times 10^{11}$ MAC operations for

the forward pass and $1.2 \times 10^{11}$ MAC operations for the weight update. Training the same layer via 100 epochs of backpropagation requires $6.0 \times 10^{12}$ MAC operations for forward passes and $1.8 \times 10^{13}$ operations for weight updates. Forward projection requires only a single training epoch, reducing overall computation time accordingly. Note that Local Supervision and Forward-Forward algorithms have similar computational and memory requirements as backpropagation (Table 2 and Table S3). We also present the wall-clock time for full training of an

**Table 2 | Training complexity for a single hidden layer with $m$ inputs and m outputs, given a label with dimension $m_L$**

| Method | Compute | | | Memory | | FMNIST training | |
|---|---|---|---|---|---|---|---|
| | Training epochs | Forward pass | Weight update | Model parameters | Weight update | Time (s) | Epochs |
| Backprop-agation | $N_e$ | $N_e N m^2$ | $3 N_e N m^2$ | $m^2$ | $m^2 + 2m$ | $22.43 \pm 4.8$ | $11.6 \pm 2.7$ |
| Local Supervision | $N_e$ | $N_e N (m^2 + m m_L)$ | $2 N_e N (m^2 + m m_L)$ | $m^2 + m m_L$ | $m^2 + 2m$ | $46.9 \pm 12.5$ | $10.8 \pm 3.56$ |
| Forward-Forward | $N_e$ | $2 N_e N m^2$ | $4 N_e N m^2$ | $m^2$ | $m^2 + 2m$ | $172 \pm 56$ | $40.2 \pm 13.8$ |
| Predictive Coding | $N_e$ | $N_e N m^2$ | $3 N_e N m^2$ | $2m^2$ | $2m^2 + 2m$ | $150 \pm 61$ | $25.4 \pm 10.2$ |
| Difference Target Prop. | $N_e$ | $N_e N m^2$ | $8 N_e N m^2$ | $2m^2$ | $2m^2 + 4m$ | $72.7 \pm 26.4$ | $13.6 \pm 5.3$ |
| Forward Proj. (Ours) | 1 | $N (2m^2 + m m_L)$ | $2 N m^2 + 2m^3$ | $2m^2 + m m_L$ | $2m^2$ | $0.34 \pm 0.1$ | 1 |

$N$: training sample size. $N_e$: epochs, all parameters except activations are per layer. For simplicity, we assume batch size $B = 1$ and that predictive coding trains for $k = 1$ iteration. Complexity is detailed in section S.13 in SI.

MLP with $3 \times 1000$ hidden neurons on FMNIST, demonstrating a $66 \times$ speedup with Forward Projection in this example. Further timing results are provided in Supplementary Table S3. Computations were run using the Google Colab service with an NVIDIA T4 graphics processing unit. Training complexities of other methods are discussed in section S.13 in SI.

## Discussion

We present the Forward Projection algorithm, which enables learning in a single pass over the dataset using random projections and closed-form optimisation. Compared to state-of-the-art local learning methods that require observing post-synaptic neuronal outputs to optimise synaptic weights via error-based gradient descent, Forward Projection operates under a stricter constraint, fitting weights using just the pre-synaptic neuronal activity and labels.

The target generation function proposed here for Forward Projection promotes input and label encoding in neural membrane potentials (see section S.8 in SI). Joint encoding of labels with pre-synaptic activity alleviates the degenerate neural activity which results from local modelling of simple label projections, whilst avoiding the introduction of uninformative noise (see section S.6 in SI). In this analysis, a simple non-linearity was employed for target generation. The utility of more complex nonlinearities for target generation is a subject for further research.

Explainability of neural network models is an important limitation in decision-critical fields such as biomedicine, where errors such as confounded decisions may lead to significant consequences[35]. FP-trained layers may be interpreted without downstream information, providing insight into model reasoning in hidden layers. FP interpretation yielded informative outputs in three model architectures, identifying clinically salient features in ECG sequences and OCT images. An important advantage of FP training is that saliency maps may be generated before downstream fitting, permitting on-the-fly inspection of intermediate layers for sufficiency or confounding. Thus, expert scrutiny of hidden layer performance may be conducted even before the downstream architecture is finalised.

Few-shot learning is ubiquitous in biological systems, which exhibit rapid neuronal adaptation to changing environments[36]. In our experiments, Forward Projection demonstrated clear performance advantages over backpropagation and local learning approaches in few-shot learning tasks, presenting a plausible method for learning new tasks rapidly. In this setting, convolutional features learned by Forward Projection yielded more generalisable models than backpropagation, which overfitted in some experiments and underfitted in others, even underperforming random features in some cases. Forward Projection also maintained reasonable performance on activation functions that were untrainable by SGD-based methods (see section S.14 in SI).

The closed-form Forward Projection fit is computable in a single pass over the data for each layer, presenting an opportunity to expedite training and reduce environmental footprint. The efficiency of FP training is attributable to a substantially different operational sequence from that employed in SGD-based approaches. Firstly, FP training collects a Gram matrix of pre-synaptic activity over a single epoch, which is subsequently inverted during a one-step weight matrix computation. Secondly, FP completes fitting for each layer before initialising successive layers. In contrast, SGD-based approaches fit all the layers in the network iteratively, using feedback from neuronal outputs and downstream layers. FP training requires no retrograde communication between neural output activations and presynaptic connections, enabling direct training on hardware with unidirectional synaptic and neuronal communication. This differs from current iterative local learning methods, which require backward communication from neuronal outputs to optimise pre-synaptic parameters (see section S.2 in SI). Although FP addresses the feedback-free constraint inherent in biological learning systems, the biological plausibility of one-step learning via matrix inversion remains uncertain. Iterative fitting via gradient descent in sequential, batch-based learning offers a practical and scalable approach for aligning weights with locally generated targets, as defined in (1) (see section S.11 in SI). Although closed-form optimisation remains a challenge for recurrent architectures, future research will explore integrating iterative training with random temporal convolutions to generalise the FP framework to dynamic models and expand its applicability to temporal learning tasks. This backpropagation-free strategy holds promise for biologically inspired computing systems, such as spiking neural networks, where non-differentiable activation functions preclude standard backpropagation[37–39]. FP training may be applicable as a pre-training step to reduce the number of training epochs required for backpropagation. In conclusion, FP is an efficient approach for neural network optimisation, employing techniques from randomised projective embedding and linear regression to fit weight matrices in one epoch using a single-step solution. Interpretability of hidden neurons in FP-trained models may be employed to improve the explainability of neural network predictions.

## Methods

Generalisability of machine learning methods to real-world datasets requires robustness to adverse modelling conditions such as class imbalance and noise, which often impede performance[40]. To assess the applicability and generalisability of Forward Projection and local learning methods in diverse real-world conditions, performance was evaluated in benchmark tasks from four biomedical domains described below.

### PTBXL-MI

The PTB-XL dataset contains 12-lead electrocardiography (ECG) recordings from 18,889 participants[29,41]. ECG recordings of ten-second duration and 100Hz sample rate were used in our experiments. The predictive task was to discriminate ECG recordings with normal

waveform morphology ($N_{\text{train}}$ = 6, 451; $N_{\text{test}}$ = 721) from those diagnosed as myocardial infarction ($N_{\text{train}}$ = 2, 707; $N_{\text{test}}$ = 268) by a cardiologist. Data instances with uncertain diagnoses were excluded from this analysis. Following recommendations of the dataset authors who provided predefined participant-disjoint dataset splits, the tenth fold was held out for model testing[29]. The first 15 MI-positive ECGs in the test set were annotated by a medical doctor to identify diagnostically relevant sections for quantitative explainability evaluation.

### Promoters

The Human Non-TATA Promoters dataset ("Promoters") was extracted from the GenomicBenchmarks repository[28]. Data was originally published in ref. 27. 36,131 nucleotide sequences of 251 bases each were analysed. Nucleotide sequences were converted to 4-channel one-hot vectors indicating adenine, cytosine, guanine and thymine. Indeterminate bases were represented with zero vectors. Models were required to classify the promoter functionality of the sequence as "promoter" ($N_{\text{train}}$ = 12,355; $N_{\text{test}}$ = 4119) or "non-promoter" ($N_{\text{train}}$ = 14, 742; $N_{\text{test}}$ = 4915).

### CXR

The paediatric pneumonia chest X-ray dataset ("CXR") is a retrospective cohort of patients aged between one and five years, recorded in Guangzhou Women and Children's Medical Center, Guangzhou, originally published in ref. 31. Chest X-ray images were recorded as part of routine care during diagnostic workup for suspected lower respiratory tract infection. During data collection, clinicians screened the images for quality and excluded those with severe artefacts or corruption. Images were annotated by two expert physicians. Local institutional review board approvals were obtained. Images were loaded in greyscale, rescaled to the [0, 1] intensity range and resized to 128 × 128 pixels by bilinear interpolation. Models were required to classify images as "normal" ($N_{\text{train}}$ = 1349; $N_{\text{test}}$ = 234), "viral pneumonia" ($N_{\text{train}}$ = 2538; $N_{\text{test}}$ = 242), or "bacterial pneumonia" ($N_{\text{train}}$ = 1345; $N_{\text{test}}$ = 148).

### OCT

The optical coherence tomography dataset ("OCT") is a retrospective cohort of adult patients from five ophthalmology institutions in the USA and China recorded between 2013 and 2017 during routine care, originally published in ref. 31. Images were initially annotated by local medical students, who had received OCT interpretation training. Subsequent annotation was performed by four ophthalmologists and two independent retinal specialists. Horizontal foveal cut images were available in portable network graphics image format. Images were loaded in greyscale, rescaled to the [0, 1] intensity range, and resized to 128 × 128 pixels by bilinear interpolation. Models were required to classify images as either "normal" ($N_{\text{train}}$ = 2926; $N_{\text{test}}$ = 149) or "choroid neovascularisation" ($N_{\text{train}}$ = 791; $N_{\text{test}}$ = 178).

### Model training

The FMNIST dataset was modelled using an MLP with 3 × 1000 hidden ReLU-activated neurons. Sequential datasets (PTBXL-MI and Promoters) were modelled by a 1D-CNN architecture of four convolutional blocks. Each convolutional block included two convolutional layers with kernel dimensions 3 and strides of 1 and 2, respectively. Convolutional layers in the $l$-th block had $32 \times 2^{l-1}$ filters. Convolutional implementation is detailed in Supplementary section S.3. Convolutional outputs were aggregated by global average pooling in the penultimate layer. For gradient-descent-based learning algorithms, batch normalisation layers were included between convolutional blocks. CIFAR2 modelling in Table 1 used a vision transformer architecture[42] operating on image patches of dimension 4 × 4, with a sequential stack of four multi-headed attention layers, each having 8

heads, embedding dimension 64, and MLP dimension 64. 2D-CNN architectures had kernel dimension 3 × 3 and the $l$-th convolutional block had $16 \times 2^{l-1}$ filters (CXR and OCT), $32 \times 2^{l-1}$ filters (CIFAR2), or $64 \times 2^{l-1}$ filters (CIFAR10). The sign function was employed to generate target activations for Forward Projection models, such that

$$\tilde{z}_l = \text{sign}(a_{l-1}Q_l) + \text{sign}(yU_l). \tag{4}$$

Data augmentation was not performed in our experiments. Models were fitted using the PyTorch library. For SGD-based methods, early stopping was performed according to validation loss. Model weight initialisation was random, and optimisation was performed with the Adam optimiser with a learning rate of 0.001, training to minimise validation loss by early stopping with a patience of five epochs. In few-shot experiments, the learning rate was reduced to 0.0001 for SGD-based models, and a patience of ten epochs was employed. It is acknowledged that direct comparison of gradient-descent-based training with closed-form solutions is not strictly "like-for-like". However, all efforts were made to ensure experimental conditions were otherwise equivalent. Models were fitted to minimise categorical or binary cross-entropy as appropriate. Implementations of Forward-Forward, Local Supervision, Difference Target Propagation and Predictive Coding methods are detailed in supplementary section S.5.

### Explainability analysis

Model architectures for explainability analysis were equivalent to those used for the main experiments. Pre-activation potentials in hidden neurons were interpreted as local label predictions using (3) with the following approximation

$$\hat{y}_l : = \tanh(z_l - \text{sign}(a_{l-1}Q_l))U_l^+. \tag{5}$$

Here, $\tanh(\cdot)$ is employed as a surrogate inverse for the sign function used to generate the target potentials.

### Ethics approval and consent to participate

This study involves no new data collection or experiments on human or animal subjects. All clinical data used in this study was publicly available in previous publications[27–29,31].

## Data availability

The PTBXL-MI dataset is available from PhysioNet[41]. The Promoters dataset is available from the GenomicBenchmarks library[28] and the original publication[27]. The CXR and OCT datasets are provided in ref. 31. The pre-processed datasets and cross-validation folds used in this study are available in the Mendeley Database under accession code fp_ datasets - Mendeley Data[43].

## Code availability

All code required to reproduce the study findings is available from github.com/robertoshea/forward_projection, with a digital object identifier (https://doi.org/10.5281/zenodo.17958941).

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

## Acknowledgements

Study authors are grateful to Dr. Clément Ruah for valuable discussions. This research was supported in part by the EPSRC Open Fellowship EP/X011356/1.

## Author contributions

Robert O'Shea: conceptualisation, methodology, formal analysis, software, investigation, visualisation, writing—original draft preparation. Bipin Rajendran: conceptualisation, methodology, supervision, writing—review & editing, resources, project administration, funding acquisition.

## Competing interests

Robert O'Shea contributed to the initial draft of the manuscript whilst employed at King's College London. He commenced employment with Eli Lilly & Company prior to the submission of the revised manuscript. His involvement in this project is entirely independent of his role at Eli Lilly & Company and was not influenced by his employment there. Bipin Rajendran has no competing interests to declare.
