## [Transparent Peer Review file · Nature Communications]

Closed-Form Feedback-Free Learning with Forward Projection

Corresponding Author: Dr Robert O'Shea

Version 0:

Reviewer comments:

Reviewer #1

(Remarks to the Author)

The motivation for this paper is the well known problem of synaptic credit assignment when trying to find biologically plausible mechanisms for learning in multilayer networks, which is computationally achieved through back-propagation. The authors propose summing a nonlinear transformation of a fixed random projection of the output labels and of the inputs to a unit as the error signal guiding the unit. The error signal guides the synaptic INPUTS to the unit, i.e. before the non-linearity. Using a least square loss, and since the projections are fixed the update to the synaptic weight inputs becomes a simple regression problem that is solved analytically. So, as I understand it training requires just one forward pass through the network updating W_1 , getting the outputs of layer 1, then updating W_2 etc. I must say that this is not clearly stated anywhere in the main article how learning with mini-batches proceeds (it is explained in the supplement.)

Comments:

A. My main concern is that as this is motivated by finding biologically plausible alternatives to back propagation. The main computational step in this approach involves a massive matrix inversion, with absolutely no biological analog. The authors could consider solving the regression problem through gradient descent, which for the LS loss should ultimately converge to the regression solution, however they would have to run through the whole forward pass for each batch. Note that the gradient descent step for some losses has some similarity to a Hebbian update rule, see [1], though the LS loss has some issues in that regard.

B. An additional problem with biological plausibility is that in a multiclass case the Z's are m_L dimensional and it is unclear how such a signal could reach a single unit. That is why people have proposed layerwise learning with connections to an output layer with C units (for the C classes).

C. In fact, other than the ability to compute the weights in closed form (which as indicated above is not biologically plausible) this approach is not that different from layerwise training which can also be done sequentially from the first layer to the last. For a more biologically plausible layerwise learning see the proposed randomized layerwise learning, which chooses a random layer to update at each step. See [2].

D. It is hard for me to evaluate the success of the algorithm because other than FMNIST I am not familiar with the other data sets. Typically many methods work well on simple datasets like FMNIST and MNIST. So the challenge is to evaluate the methods on datasets like cifar10 or cifar100. Also, it would be helpful to have more precise information on the models used. Finally, it is not 'fair' to compare gradient descent algorithms with one that provides an analytic solution to each minimization.

E. A few relevant references the authors might be interested in.

[1] Yali Amit, 'Deep learning with asymmetric connections and Hebbian updates', *Frontiers in Computational Neuroscience*. (2019), DOI:10.3389/fncom.2019.00018

[2] Mufeng T., Yibo Y. and Amit, Y. (2022), Biologically Plausible Training Mechanisms for Self-Supervised Learning in Deep Networks, *Front. Comput. Neurosci.*, <https://doi.org/10.3389/fncom.2022.789253>

[3] Bernd Illing, Jean Robin Ventura, Guillaume Bellec, and Wulfram Gerstner. Local plasticity rules can learn deep representations using self-supervised contrastive predictions. In *Thirty-Fifth Conference on Neural Information Processing Systems*, 2021

(Remarks on code availability)

Reviewer #2

(Remarks to the Author)

The paper introduces Forward Projection (FP) as a biologically plausible, feedback-free training method for neural networks. FP proposes an innovative approach that eliminates the need for retrograde communication, traditionally required by backpropagation, by leveraging a single forward pass for weight optimization using closed-form regression. The authors present FP as a method capable of delivering interpretable predictions, substantial computational efficiency, and strong performance in few-shot learning scenarios, particularly within biomedical applications. While these promises are compelling, the claims lack full theoretical and empirical support, raising questions about the method's robustness, scalability, and generalizability.

Strengths and Contributions

The FP framework introduces several noteworthy innovations. First, the method eliminates backward communication during training, addressing a significant limitation in biological plausibility for traditional backpropagation. Second, applying FP to biomedical datasets, such as ECG signals and retinal OCT images, demonstrates its potential to handle real-world challenges in medical decision-making while providing interpretable insights into hidden neuron activity. Finally, the efficiency of FP is a key advantage, as it claims significant speed-ups by avoiding iterative gradient descent. This is especially attractive for tasks with large-scale datasets or limited computational resources.

Weaknesses and Limitations

Despite its novel contributions, the paper has notable weaknesses that limit its impact.

Lack of Theoretical Rigor

The paper does not adequately address the theoretical underpinnings of FP. For instance, it remains unclear whether closed-form regression can reliably converge to optimal solutions for deep networks with complex non-linearities. The stability of FP under scenarios involving noisy or imbalanced datasets is also unexamined. Additionally, the trade-offs introduced by the random projection matrices (QI and UI), which may significantly affect the reliability of target generation, are not analyzed.

Insufficient Empirical Validation

While FP is compared to a few local learning methods, such as Forward-Forward and local supervision, the experimental scope is narrow. The paper lacks comparisons with state-of-the-art biologically plausible approaches like predictive coding, energy-based models, and Difference Target Propagation. Furthermore, the empirical analysis does not include ablation studies to assess the role of critical components like the choice of random projections or the regularization parameter (λ) in ridge regression. The experiments are confined to biomedical datasets and relatively simple architectures (MLPs and CNNs), leaving questions about the method's scalability and performance on more diverse datasets and architectures (e.g., transformers).

Generalization and Interpretability

The paper claims that FP generalizes well in few-shot learning tasks. However, it does not investigate the underlying reasons for this performance. There is no exploration of FP's robustness under adversarial conditions or noisy data scenarios. Additionally, while the authors emphasize the interpretability of FP-trained models, no formal metrics (e.g., faithfulness or consistency) are used to validate these claims. A comparison with popular interpretability techniques, such as saliency maps or SHAP, would strengthen the argument for FP's unique benefits in decision-critical domains like healthcare.

Suggestions for Improvement

To enhance the impact and validity of the proposed method, the following improvements are recommended:

1. Comprehensive Benchmarks:
 - o Include comparisons with both biologically inspired approaches (e.g., predictive coding, Difference Target Propagation) and traditional backpropagation methods for efficiency and accuracy.
2. Theoretical Analysis:
 - o Provide convergence guarantees and investigate the sensitivity of FP to random projection matrices and regularization parameters.
3. Expanded Experiments:
 - o Evaluate FP on larger, more diverse datasets (e.g., ImageNet, CIFAR-10) and architectures (e.g., transformers, autoencoders) to establish its scalability and robustness.
4. Interpretability Metrics:
 - o Use quantitative metrics to validate interpretability claims and compare FP's interpretability with backpropagation-trained models using established explainability techniques.
5. Ablation Studies:
 - o Assess the contribution of key components such as random projections and ridge regression to the overall performance.

Related Works to Cite

The paper would benefit from citing the following works to situate FP within the broader context of biologically inspired and feedback-free learning methods such as "Forward-Forward Algorithm: Hinton, G. E. (2022). The forward-forward algorithm: Some preliminary investigations. arXiv:2212.13345"

The paper introduces an exciting framework with the potential to advance biologically inspired machine learning. However, the claims made about FP's efficiency, interpretability, and generalization are not fully substantiated through theoretical or empirical analysis. Addressing the outlined weaknesses and incorporating comparisons with existing approaches could significantly enhance the paper's credibility and impact. A major review is required to validate the claims and strengthen the contributions of this work.

(Remarks on code availability)

No code was available; the provided GitHub repo associated with the provided link was either private or nonexistent.

Reviewer #3

(Remarks to the Author)

This manuscript presents a novel approach, Forward Projection (FP), for backpropagation-free learning, which aims to address the challenge of pre-synaptic weight optimization in the absence of retrograde communication. The method introduces a randomized closed-form training technique that requires only a single forward pass over the entire dataset, thereby making it computationally efficient. The study demonstrates the effectiveness of FP across multiple biomedical datasets, showing its ability to generate interpretable neural network models without feedback from neuronal outputs or downstream layers. The method's interpretability is particularly intriguing, as it successfully identifies clinically relevant features in electrocardiogram and optical coherence tomography datasets.

While the paper presents an interesting and novel approach, there are several areas that could benefit from further clarification and additional experiments:

1. The interpretable nature of FP, particularly the identification of clinically important features like myocardial infarction signatures and choroid neovascularisation detection, is indeed very interesting. This is a key strength of the paper. And it would be valuable to see a more in-depth analysis of the interpretability, including how the generated target values and layer-wise information are linked to specific clinical outcomes, potentially with additional examples or visualizations to enhance the readers' understanding.
2. The paper would benefit from more details on the types of network architectures that the FP method applies to. Specifically, does FP generalize to more complex and widely used architectures like ResNet or Transformer-based models? Clarifying this would help establish the broader applicability of the method.
3. The paper compares FP with methods like forward-forward training and local supervision. However, it would be beneficial to include a comparison with other local learning methods, such as Hebbian learning, which may be relevant to the proposed method. Providing a detailed comparison of these methods on the datasets used in the study, including accuracy, training time, and convergence speed, would add depth to the evaluation.
4. The manuscript demonstrates FP on the FMNIST dataset, but it would be helpful to extend the validation to larger visual datasets, such as CIFAR-100 or ImageNet. This would allow the authors to assess the scalability of the FP method and compare its performance against other state-of-the-art approaches in vision tasks.
5. It would be useful to include more detailed statistics on training metrics, such as training time, memory consumption, and convergence speed, especially for the biomedical datasets. These statistics would provide a clearer picture of the computational efficiency of FP in comparison to other methods, particularly in terms of real-world applicability.
6. The FP method presented in this paper shares some similarities with the concept of decoupled learning (both forward and backward decoupling, such as study in [1]Czarnecki, W. M., Świrszcz, G., Jaderberg, M., Osindero, S., Vinyals, O., & Kavukcuoglu, K. (2017, July). Understanding synthetic gradients and decoupled neural interfaces. In International Conference on Machine Learning (pp. 904-912). PMLR.), which has been proposed to address issues in traditional backpropagation. A discussion of how FP differs from or relates to decoupled learning approaches would help readers understand the novelty and advantages of FP in the context of existing research in this area.

Therefore, the proposed FP method is an exciting and innovative approach to backpropagation-free learning, offering significant potential in biomedical applications and other domains. To strengthen the paper and provide a more comprehensive evaluation, it would be important to address the points above

(Remarks on code availability)

The README file could provide more detailed running examples.

Version 1:

Reviewer comments:

Reviewer #1

(Remarks to the Author)

The manuscript has improved significantly with the new experiments and supplemental analysis.

My only concern is that the authors seemed to have avoided the multi-label setting in their experiments because according to their own analysis it doesn't perform well. That seems to be a fundamental drawback. For example they do not show results on the standard MNIST ten class data set or 10 classes in cifar10.

One way they may try to resolve this is by doing what SVM methods do: K one class against the rest classifiers which then vote.

I think the paper can be accepted with at least a more detailed investigation into the multiclass case.

(Remarks on code availability)

Reviewer #3

(Remarks to the Author)

Most of the concerns raised in the previous round have been adequately addressed, and the manuscript has improved substantially. However, during a careful reading, I noticed that the proposed method also demonstrates potential advantages in terms of energy efficiency, which could make the work more relevant to the broader neuromorphic computing and spiking neural networks (SNNs) community. I would therefore encourage the authors to add a brief discussion, either in the Related Works or Discussion section, about the connections and distinctions between the proposed Forward Projection framework and SNNs.

The recent advances in energy-efficient or structurally novel SNNs could be referred, such as [1] J. Shen et al., Efficient Spiking Neural Networks with Sparse Selective Activation for Continual Learning, AAAI 2024; [2] X. Shi et al., SpikingResformer: Bridging ResNet and Vision Transformer in Spiking Neural Networks, arXiv:2403.14302; and [3] Shen et al., Spiking neural networks with temporal attention-guided adaptive fusion for imbalanced multi-modal learning, arXiv:2505.14535.

(Remarks on code availability)

There is no problem.

Version 2:

Reviewer comments:

Reviewer #1

(Remarks to the Author)

Authors have addressed my concerns.

(Remarks on code availability)

Reviewer #3

(Remarks to the Author)

Thanks for the responses. All my concerns have been resolved, and I recommend accepting this paper.

(Remarks on code availability)

There is no problem about the code.

Rebuttal to Reviewer #1

Reviewer Comment: *The motivation for this paper is the well-known problem of synaptic credit assignment when trying to find biologically plausible mechanisms for learning in multilayer networks, which is computationally achieved through back-propagation. The authors propose summing a nonlinear transformation of a fixed random projection of the output labels and of the inputs to a unit as the error signal guiding the unit. The error signal guides the synaptic INPUTS to the unit, i.e. before the non-linearity. Using a least square loss, and since the projections are fixed the update to the synaptic weight inputs becomes a simple regression problem that is solved analytically. So, as I understand it training requires just one forward pass through the network updating W_1 , getting the outputs of layer 1, then updating W_2 etc. I must say that this is not clearly stated anywhere in the main article how learning with mini-batches proceeds (it is explained in the supplement.)*

Response: Thank you for your time reviewing this manuscript, and for your thoughtful comments. We believe that the revisions made in response to your suggestions have substantially strengthened the manuscript.

Reviewer Comment: *My main concern is that as this is motivated by finding biologically plausible alternatives to back propagation. The main computational step in this approach involves a massive matrix inversion, with absolutely no biological analog. The authors could consider solving the regression problem through gradient descent, which for the LS loss should ultimately converge to the regression solution, however they would have to run through the whole forward pass for each batch. Note that the gradient descent step for some losses has some similarity to a Hebbian update rule, see [1], though the LS loss has some issues in that regard.*

Response: We appreciate the reviewer’s concern regarding the biological plausibility of our method, particularly the reliance on matrix inversion. Whilst our approach is inspired by the biological constraint of forward-only signalling, we acknowledge that the closed-form solution involving matrix inversion lacks a direct biological analog. To clarify this, we have revised the manuscript to state: “*The objective of this approach is to develop a neural network training algorithm which requires no retrograde communica-*

tion.” We did consider optimizing the forward projection (FP) local loss via gradient descent. However, as noted, this would require a full forward pass for each batch and layer-wise multi-epoch training, resulting in significant computational overhead. Moreover, gradient descent inherently involves error feedback from post-synaptic to pre-synaptic weights, which introduces retrograde communication—contrary to our core constraint.

Reviewer Comment: *B. An additional problem with biological plausibility is that in a multiclass case the Z 's are m_L dimensional and it is unclear how such a signal could reach a single unit. That is why people have proposed layerwise learning with connections to an output layer with C units (for the C classes).*

Response: We agree that learning high-dimensional labels using individual neurons presents a key challenge biological plausibility. To examine this, we have added a further technical analysis section exploring the effect of label dimension on FP representation learning in hidden layers in Section 1.10. In our approach, random projection is employed to map the multidimensional signal into a real-valued univariate signal at each neuron level. Whilst this may be lossy at the neuron level, the collective activity across neurons retains sufficient structure to enable approximate recovery of class-specific predictions. This is achieved by aggregating neuron outputs using the Moore-Penrose inverse of the predefined label projection matrix, as described in Equation 3. Figure A2-A, illustrates this process, showing how multiclass FMNIST predictions can be reconstructed from hidden layer activations.

Reviewer Comment: *In fact, other than the ability to compute the weights in closed form (which as indicated above is not biologically plausible) this approach is not that different from layerwise training which can also be done sequentially from the first layer to the last. For a more biologically plausible layerwise learning see the proposed randomized layerwise learning, which chooses a random layer to update at each step.*

Response: We appreciate the reviewer’s observation regarding the similarities between our method and layerwise training. Indeed, our approach shares structural features with sequential layerwise learning. However, our contributions are twofold:

1. Elimination of retrograde communication: Our method removes the need for any backward signaling between layers, a key departure from conventional layerwise training.

2. Closed-form weight computation: We introduce a closed-form solution that enables forward-only training without iterative updates, which distinguishes our approach from gradient-based methods.

Additionally, our method differs from local supervision in a fundamental way. Rather than encoding only label information, we aim to encode both label and pre-synaptic activity within hidden neuron potentials. As discussed in Remark 1.5 of the Supplementary Information, simple label projection tends to produce highly correlated hidden representations, with rank constrained by label dimensionality. Our forward projection mechanism addresses this by decorrelating label signals through informative data projections.

This distinction becomes particularly evident in few-shot learning scenarios (Figure 2), where forward projection avoids label overfitting by jointly optimizing label and input representations. We have acknowledged the limitations in biological plausibility—particularly the use of matrix inversion—and clarified that our primary motivation is to explore closed-form learning strategies that operate without retrograde communication.

Reviewer Comment: *D. It is hard for me to evaluate the success of the algorithm because other than FMNIST I am not familiar with the other data sets. Typically many methods work well on simple datasets like FMNIST and MNIST. So the challenge is to evaluate the methods on datasets like cifar10 or cifar100. Also, it would be helpful to have more precise information on the models used. Finally, it is not ‘fair’ to compare gradient descent algorithms with one that provides an analytic solution to each minimization.*

Response: Thank you for this suggestion. In response, we have now added experiments on the CIFAR10 dataset, and provided further information on the employed models. We have also added two established biologically inspired training techniques – difference target projection (DTP) and predictive coding (PC). Table 1 provides test performance of modelling using DTP and PC, with forward projection outperforming both DTP and PC on FMNIST, Promoters and PTBXL-MI datasets. Furthermore, Figure A3 and Table 4 present few-shot learning results across Promoters, PTBXL-MI and CIFAR datasets, where forward projection consistently achieves superior performance. We agree that comparison of gradient descent training

with closed-form solutions is not strictly like-for-like. However, given that gradient descent remains the standard training method for many models, we felt it was important to include it for context. To clarify this for readers, we have added the following note to the Methods section: "It is acknowledged that direct comparison of gradient descent based training with closed-form solutions is not an "like-for-like" comparison; however all possible efforts were made to ensure experimental conditions were otherwise equivalent."

Reviewer Comment: *A few relevant references the authors might be interested in:*

1. Yali Amit, 'Deep learning with asymmetric connections and Hebbian updates', *Frontiers in Computational Neuroscience*. (2019),
2. Mufeng T., Yibo Y. and Amit, Y. (2022), *Biologically Plausible Training Mechanisms for Self-Supervised Learning in Deep Networks*, *Front. Comput. Neurosci.*,
3. Bernd Illing, Jean Robin Ventura, Guillaume Bellec, and Wulfram Gerstner. *Local plasticity rules can learn deep representations using self-supervised contrastive predictions*. In *Thirty-Fifth Conference on Neural Information Processing Systems, 2021*

Response: Thank you for highlighting these manuscripts. We found these to be relevant as context for our method. We have discussed Amit2019 and Tang2022 in the introductory section on background and related work

Rebuttal to Reviewer #2

Reviewer Comment: *The paper introduces Forward Projection (FP) as a biologically plausible, feedback-free training method for neural networks. FP proposes an innovative approach that eliminates the need for retrograde communication, traditionally required by backpropagation, by leveraging a single forward pass for weight optimization using closed-form regression. The authors present FP as a method capable of delivering interpretable predictions, substantial computational efficiency, and strong performance in few-shot learning scenarios, particularly within biomedical applications. While these promises are compelling, the claims lack full theoretical and empirical support, raising questions about the method's robustness, scalability, and generalizability.*

Strengths and Contributions: The FP framework introduces several noteworthy innovations. First, the method eliminates backward communication during training, addressing a significant limitation in biological plausibility for traditional backpropagation. Second, applying FP to biomedical datasets, such as ECG signals and retinal OCT images, demonstrates its potential to handle real-world challenges in medical decision-making while providing interpretable insights into hidden neuron activity. Finally, the efficiency of FP is a key advantage, as it claims significant speed-ups by avoiding iterative gradient descent. This is especially attractive for tasks with large-scale datasets or limited computational resources.

Response: Thank you for your time reviewing this manuscript, and your thoughtful comments. We address the specific points below:

Reviewer Comment: *Lack of theoretical guarantees for convergence and stability under noise.*

Response: Thank you for raising this important point. We have added a new theoretical analysis (Section 1.9) examining convergence behavior in a setting where both input data and labels are pure Gaussian noise. We have also added further theoretical analysis on the impact of label dimension on representation learning in hidden layers 1.10. To further explore robustness, we have conducted experiments where Gaussian noise is injected into the input data. As summarized in Figure A2-C, forward projection maintains strong performance under substantial noise, achieving FMNIST test accuracy above 0.819 even at signal-to-noise ratios as low as 1:4.6.

Reviewer Comment: *No analysis of the effects of random projections or ridge regression parameter λ .*

Response: Thank you for highlighting this important point. We have now included re-randomisation experiments study to assess the sensitivity of our method to the choice of projection matrices. As shown in Figure A2-D, model performance on FMNIST remains stable across different randomisations, suggesting robustness to projection variability. Additionally, we have conducted experiments varying the ridge regression penalty parameter λ , as presented in (Figure A2-B). Stable performance was observed varying the ridge regression penalty from $\lambda = 1.25$ to $\lambda = 80$, with the highest

performance at $\lambda = 80$.

Reviewer Comment: *Empirical scope is narrow, with limited model variety.*

Response: Thank you for this valuable feedback. We have added an implementation of the forward projection method for multi-head attention layers 1.3. Thus, forward projection is implemented in a vision transformer architecture on the CIFAR dataset (Table 1). Additionally, we have expanded our methodological comparisons to include two biologically inspired training approaches—Predictive Coding (PC) and Difference Target Propagation (DTP)—to provide a more comprehensive benchmark. These additions help contextualize the performance and generalizability of forward projection across both architectures and learning paradigms.

Reviewer Comment: *No formal metrics were used to validate interpretability claims.*

Response: Thank you for highlighting the need for formal validation of interpretability claims. In response, we have added a quantitative evaluation of the interpretability of forward projection layer explanations using the PTBXL-MI dataset (Section). Fifteen electrocardiograms indicating myocardial infarction were annotated by a medical doctor with MRCP(UK) certification and nine years of cardiology experience. Diagnostic regions were manually labeled, and we assessed the discriminative power of the layer explanations using AUC and AUPR metrics. For comparison, we applied Gradient-weighted Class Activation Mapping (GradCAM) to networks trained via backpropagation. Explanations derived from the sixth convolutional layer of the forward projection model (\hat{y}_6), using Equationeq:layer explanation, achieved comparable performance to GradCAM:

(a) Forward Projection: $\text{AUC} = 0.61 \pm 0.14$, $\text{AUPR} = 0.52 \pm 0.19$

(b) GradCAM: $\text{AUC} = 0.59 \pm 0.12$, $\text{AUPR} = 0.55 \pm 0.13$.

We have also included a visual comparison of the layer explanations in Figure 3-B, illustrating the qualitative similarity between the two approaches.

Reviewer Comment: *Code repository was inaccessible.*

Response: We apologize for this inconvenience. Unfortunately, at the time of writing, our allotted code ocean computational resources have been exhausted. We have requested additional compute resources. We have provided reproducible sample experiment scripts at the provided github link. Thank you for your thoughtful and constructive feedback. We believe that the revisions made in response to your suggestions have substantially strengthened the manuscript.

Rebuttal to Reviewer #3

Reviewer Comment: *Would benefit from more in-depth interpretability analysis and visualizations.*

Response: Thank you for this insightful suggestion. To strengthen our interpretability analysis, we have added both quantitative metrics and visual comparisons to evaluate the explanatory power of forward projection layer outputs.

Specifically, we have added a quantitative evaluation of the interpretability of forward projection layer explanations using the PTBXL-MI dataset (Section). Fifteen electrocardiograms indicating myocardial infarction were annotated by a medical doctor with MRCP(UK) certification and nine years of cardiology experience. Diagnostic regions were manually labeled, and we assessed the discriminative power of the layer explanations using AUC and AUPR metrics.

For comparison, we applied Gradient-weighted Class Activation Mapping (GradCAM) to networks trained via backpropagation. Explanations derived from the sixth convolutional layer of the forward projection model (\hat{y}_6), using Equationeq:layer explanation, achieved comparable performance to GradCAM:

(a) Forward Projection: $AUC = 0.61 \pm 0.14$, $AUPR = 0.52 \pm 0.19$

(b) GradCAM: $AUC = 0.59 \pm 0.12$, $AUPR = 0.55 \pm 0.13$

To complement these metrics, we have added Figure 3-B, which visually compares the layer explanations produced by forward projection and GradCAM. These visualizations highlight the qualitative similarity in the regions identified as diagnostically relevant.

Reviewer Comment: *Unclear whether FP generalizes to architectures like*

ResNet or Transformers.

Response: Thank you for raising this important point. To evaluate the generalizability of forward projection beyond simple feedforward architectures, we have implemented the method within multi-head attention layers, enabling its use in a Vision Transformer (ViT) architecture (remark1.3 in supplementary information). This implementation is tested on the CIFAR dataset (Table 1), demonstrating that forward projection can be successfully applied to transformer-based models. This extension supports the broader applicability of the method and highlights its potential for integration into modern deep learning architectures.

Reviewer Comment: *Consider comparison to Hebbian learning and decoupled learning approaches.*

Response: Thank you for this suggestion. We have now included comparisons with two biologically inspired training methods—Predictive Coding (PC) and Difference Target Propagation (DTP)—which share conceptual similarities with Hebbian and decoupled learning approaches. These methods were selected due to their relevance in the literature and their alignment with the goals of biologically plausible learning.

Forward projection was benchmarked against PC and DTP across multiple datasets, including FMNIST, Promoters, PTBXL-MI, and CIFAR-10. As shown in Table1 and Figure A3, forward projection consistently outperformed both methods in standard and few-shot learning settings.

Reviewer Comment: *Extend validation to datasets beyond FMNIST.*

Response: Thank you for this suggestion. In addition to our original experiments on FMNIST, CXR, PTBXL-MI, and Promoters datasets, we have now extended our evaluation to include the CIFAR-10 dataset. This addition provides a more challenging benchmark and helps demonstrate the generalizability of the forward projection method across a broader range of data types and domains.

Reviewer Comment: *Include training time, memory, and convergence statistics.*

Response: Thank you for this suggestion. We have now included training time and epoch statistics for forward projection across multiple architectures—MLP, 1D-CNN, and Vision Transformer—on FMNIST, PTBXL-MI, Promoters, and CIFAR-10 datasets. These results are presented in Table??, providing a clearer picture of the computational efficiency and convergence behavior of the method.

Reviewer Comment: *README lacks sufficient usage examples.*

Response: Thank you for pointing this out. We have revised the README to include a clear usage example along with more detailed instructions for running the code.

Thank you for your thoughtful and constructive feedback. We believe that the revisions made in response to your suggestions have substantially strengthened the manuscript.

Rebuttal to Reviewer #1

Reviewer Comment: *The manuscript has improved significantly with the new experiments and supplemental analysis. My only concern is that the authors seemed to have avoided the multi-label setting in their experiments because according to their own analysis it doesn't perform well. That seems to be a fundamental drawback. For example they do not show results on the standard MNIST ten class data set or 10 classes in cifar10. One way they may try to resolve this is by doing what SVM methods do: K one class against the rest classifiers which then vote. I think the paper can be accepted with at least a more detailed investigation into the multiclass case.*

Response: We sincerely appreciate your thoughtful feedback and your recognition of the improvements made to the manuscript. In response to your concern regarding the multiclass setting, we have now included results from 10-class classification experiments on CIFAR10. Additionally, the manuscript presents results on other multiclass datasets, including FMNIST (10 classes) and PTBXL-MI (3 classes). Regarding your suggestion to adopt a one-vs-all strategy similar to SVMs: our method inherently aligns with this paradigm. Since the output layer neurons are trained via linear regression, each class is effectively modeled independently, which mirrors the one-vs-all framework. While we considered implementing explicit K one-vs-all classifiers to address class imbalance, we opted for a unified model to maintain consistency with comparative baselines and ensure fair evaluation. We thank you again for your valuable insights and constructive suggestions, which have helped strengthen the manuscript.

Rebuttal to Reviewer #3

Reviewer Comment: *Most of the concerns raised in the previous round have been adequately addressed, and the manuscript has improved substantially. However, during a careful reading, I noticed that the proposed method also demonstrates potential advantages in terms of energy efficiency, which could make the work more relevant to the broader neuromorphic computing and spiking neural networks (SNNs) community. I would therefore encourage the authors to add a brief discussion, either in the Related Works or Discussion section, about the connections and distinctions between the proposed Forward Projection framework and SNNs. The recent advances in*

energy-efficient or structurally novel SNNs could be referred, such as [1] J. Shen et al., *Efficient Spiking Neural Networks with Sparse Selective Activation for Continual Learning*, AAAI 2024; [2] X. Shi et al., *SpikingRes-former: Bridging ResNet and Vision Transformer in Spiking Neural Networks*, arXiv:2403.14302; and [3] Shen et al., *Spiking neural networks with temporal attention-guided adaptive fusion for imbalanced multi-modal learning*, arXiv:2505.14535.

Response: We thank you for your thoughtful observation and valuable suggestion. We agree that the potential energy efficiency of our proposed Forward Projection framework presents a compelling connection to the broader neuromorphic computing and spiking neural networks (SNNs) community. In particular, the local learning paradigm and avoidance of backpropagation align well with principles commonly explored in SNN research. In response, we have added a brief section to the discussion to contextualize our method within ongoing efforts toward energy-efficient and structurally novel SNN architectures. The following has been added to the discussion: **Iterative fitting via gradient descent in sequential, batch-based learning offers a practical and scalable approach for aligning weights with locally generated targets, as defined in Equation 1 (see remark A10 in SI).** Although closed-form optimisation remains a challenge for recurrent architectures, future research will explore integrating iterative training with random temporal convolutions to generalise the FP framework to dynamic models and expand its applicability to temporal learning tasks. This backpropagation-free strategy holds promise for biologically inspired computing systems, such as spiking neural networks, where non-differentiable activation functions preclude standard backpropagation cite(Shen2024,Shen2025rq,Shi2024). We have added the following description of the iterative learning algorithm for Forward Projection to the SI: **An alternative to deriving the closed-form Forward Projection solution for regression over the entire dataset is to approximate the optimization procedure through iterative updates on mini-batches, while preserving local learning dynamics and target generation as formalized in Equation (1).** We partition the dataset \mathcal{D} into T mini-batches $\mathcal{B}_t = \{(\mathbf{x}_j, \mathbf{y}_j)\}_{j=1}^B$, where B is the batch size. For each mini-batch, compute $\mathbf{A}_{l-1}^{(t)} \in \mathbb{R}^{B \times m_{l-1}}$ and $\tilde{\mathbf{Z}}_l^{(t)} \in \mathbb{R}^{B \times m_l}$ using the target generation rule

$$\tilde{\mathbf{z}}_l = g_l(\mathbf{a}_{l-1}\mathbf{Q}_l) + g_l(\mathbf{y}\mathbf{U}_l). \quad (1)$$

For layer l , define the local loss:

$$\mathcal{L}_l^{(t)} = \frac{1}{B} \sum_{j=1}^B \|\mathbf{z}_{j,l} - \tilde{\mathbf{z}}_{j,l}\|^2, \quad (2)$$

where $\mathbf{z}_l^{(j)} = \mathbf{a}_{l-1}^{(j)} \mathbf{W}_l$. The gradient of $\mathcal{L}_l^{(t)}$ with respect to \mathbf{W}_l is:

$$\nabla_{\mathbf{W}_l} \mathcal{L}_l^{(t)} = \frac{2}{B} (\mathbf{A}_{l-1}^{(t)})^\top (\mathbf{A}_{l-1}^{(t)} \mathbf{W}_l - \tilde{\mathbf{Z}}_l^{(t)}). \quad (3)$$

Therefore, weights may be updated according to

$$\mathbf{W}_l \leftarrow \mathbf{W}_l - \eta \nabla_{\mathbf{W}_l} \mathcal{L}_l^{(t)}, \quad (4)$$

where η is the learning rate. Optionally, regularisation may be included via the ridge penalty:

$$\nabla_{\mathbf{W}_l} \mathcal{L}_l^{(t)} += 2\lambda \mathbf{W}_l. \quad (5)$$

In experiments on FMNIST classification by MLPs with 3×1000 hidden neurons, sequential FP training achieved similar test performance to the closed-form FP solution (test AUC: $98.6 \pm 0.79\%$, test accuracy: $84.4 \pm 0.02\%$). We appreciate your insight, which has helped us broaden the relevance and impact of our work.

Reviewer Comment: *Remarks on code availability: There is no problem.*

Response: Thank you for your time reviewing this code. We have updated the GitHub repository and Colab demo page with the code for all new experiments in the revised manuscript.